# Fault Identification of Electric Submersible Pumps Based on Unsupervised and Multi-Source Transfer Learning Integration

**Peihao Yang** [1,†], **Jiarui Chen** [1], **Lihao Wu** [2,†] **and Sheng Li** [1,*]

1   Faculty of Mathematics and Computer Science, Guangdong Ocean University, Zhanjiang 524088, China
2   School of Computer Engineering, Guangzhou City University of Technology, Guangzhou 510800, China
*   Correspondence: lish_ls@sina.com or lish_ls@gdou.edu.cn
†   These authors contributed equally to this work.

**Abstract:** The ratio between normal data and fault data generated by electric submersible pumps (ESPs) in production is prone to imbalance, and the information carried by the fault data generally as a minority sample is easily overwritten by the normal data as a majority sample, which seriously interferes with the fault identification effect. For the problem that data imbalance under different working conditions of ESPs causes the failure data to not be effectively identified, a fault identification method of ESPs based on unsupervised feature extraction integrated with migration learning was proposed. Firstly, new features were extracted from the data using multiple unsupervised methods to enhance the representational power of the data. Secondly, multiple samples of the source domain were obtained by multiple random sampling of the training set to fully train minority samples. Thirdly, the variation between the source domain and target domain was reduced by combining weighted balanced distribution adaptation (W-BDA). Finally, several basic learners were constructed and combined to integrate a stronger classifier to accomplish the ESP fault identification tasks. Compared with other fault identification methods, our method not only effectively enhances the performance of fault data features and improves the identification of a few fault data, but also copes with fault identification under different working conditions.

**Keywords:** imbalance data; fault identification; electric submersible pumps (ESPs); unsupervised; transfer learning

## 1. Introduction

Electric submersible pumps are a widely used artificial lifting tool for deep-sea oil operations [1], and their fault identification can effectively monitor the safety of production and effectively prevent the occurrence of major accidents. Carrying out fault identification of ESPs can predict or discover equipment faults in advance, so that timely measures can be taken to keep the equipment in the best condition and maintain the integrity of the equipment. It can not only avoid equipment running with faults, thus saving energy, prolonging equipment life, and making equipment function with maximum efficiency, but also, it can greatly reduce or prevent equipment accidents, thus avoiding the subsequent huge losses and bringing huge economic benefits to the enterprise.

With the development of sensor technology and data acquisition systems, various ESP data, such as pump frequency, motor temperature, and motor current, can be recorded during the production process [2]. A data-driven ESP fault identification method is implemented through training and self-learning based on normal and fault data. Fault identification is performed using the mapping relationship between fault types and data features. Many data-driven models have been developed, such as SVMs [3], ANN [4], PCA [5], and other artificial intelligence models. Liu et al. proposed a chicken flock optimization SVM model for pump fault diagnosis [6]. Chen et al. proposed an improved KNN fault detection method based on the marginal distance for pump faults [7]. Matheus et al.

proposed a random forest-based ESP data analysis method for multi-fault classification [8]. Liu et al. implemented the identification of pump fault states using XGBoost [9].

Although these methods all achieved relatively good results, they all assumed a balanced amount of faulty and normal samples in the experimental data and were limited by the need for a sufficient number of labeled training samples to learn. The same is true for ESPs due to many factors, such as the accuracy of different manufacturing processes, the use of the environment, changes in operating conditions, etc.; additionally, due to the importance of production, ESPs may be abnormal or in fault states for a year, or even years, leading us to obtain only a few samples of faults. Thus, when the ESP's failure data are limited and imbalanced in number, it becomes especially important to correctly identify the few failure samples. To identify faults in ESPs with limited samples and imbalanced distribution, the following issues are addressed: (1) how to extract the information features that are more representative of the minority fault data in the limited data, to reduce the burden on the model and increase the recognition effect of the model; (2) how to deal with the problem that the information contained in a minority of fault samples easily gets submerged in the majority of normal information, which easily leads to a high recognition error rate of the classifier for fault samples with a small amount of data.

The effectiveness of traditional machine learning methods heavily relies on data features [10], and these methods require a large amount of data to extract useful features, which is usually difficult in anomaly detection. The emergence of unsupervised learning-based anomaly detection effectively tackles this problem. Unsupervised outlier detection methods can be considered tools for extracting richer feature representations from limited data, which has also been referred to as unsupervised feature engineering [11]. This approach has been shown to be effective in enriching data representation and improving model learning [12]. Furthermore, traditional machine learning methods require the assumption that the training data and the test data obey the same data distribution, but in reality, this same distribution assumption is not always satisfied. This often requires us to relabel a large amount of training data to satisfy the training, but labeling new data is very expensive and requires a lot of human and material resources. The widespread use of transfer learning (TL) allows us to use a small amount of labeled data to mine valuable information from different working data to train the model together with a large amount of training data in different distributions.

Therefore, for the problem of the imbalance in the number ratio between normal and faulty samples in traditional machine learning-based ESP fault identification, which leads to difficulties in fault identification, we proposed an unsupervised and multi-source transfer learning integrated approach for ESP fault identification; the contributions of this paper are as follows:

(1) Combining multiple unsupervised learning methods to extract the anomaly scores (AS) generated by the unsupervised anomaly detection function as a richer representation of the data.

(2) The source domain of multiple samples was obtained by random sampling while ensuring that a minority of faulty samples were adequately selected, thus guaranteeing that a smaller number of samples could be adequately trained to improve the perception and weight of faulty samples. Then, combining the source domain training set and the target domain test set, a weak classifier based on the conditional distribution probability distribution is built for obtaining the classification results of the respective samples.

(3) The set of multiple weak classifiers is used to become a strong classifier to complete the classification recognition task.

In this paper, the experiments were conducted using real-time production data from the South China Sea oil field to prove the effectiveness and feasibility of the integrated method, which has a better performance in dealing with electric submersible pump imbalance data fault identification compared with the traditional identification method. The rest of the paper was organized as follows. In Section 2, knowledge related to the proposed

algorithm is presented. Section 3 presents the proposed algorithm. Section 4 describes the empirical evaluation method. In Section 5, the experimental results are discussed, and in Sections 6 and 7, the discussion and conclusion are presented.

## 2. Related Work

### 2.1. Unsupervised Feature Learning Methods

Unsupervised feature learning is an automatic learning of valid data features from unlabeled data. This can help subsequent machine learning models to achieve better performance more quickly [13], overcoming the limitations of supervised feature space definition. Unsupervised learning without relying on data labeling can learn the features of anomalies by different methods [14]. Commonly used unsupervised methods include model-based learning methods, distance-based learning methods, and neural network-based learning methods [15]. These learning methods have superior results and performance on the dataset when the corresponding assumptions are satisfied. In this paper, various types of unsupervised methods were used as base detectors to construct a new effective feature space.

### 2.2. Transfer Learning Method

Although traditional machine learning-based methods can have good results, without sufficient training data to support them, they often do not perform as well as they should. The widespread application of transfer learning addressed this problem well. The domain $D$ represents the subject of transfer learning, which represents the features $X$ of the data and the distribution $P(X)$ of the features. The domain is divided into target domain and source domain. The source domain $D_s$ refers to the existing knowledge domain, and the target domain $D_t$ refers to the area to be studied. Transfer learning was defined as a learning method that acquires knowledge from the source domain $D_s$ and the corresponding learning task $T_s$ [16], which is used to assist in improving the ability of the prediction function in the target domain. As shown in Figure 1, this is a schematic diagram of transfer learning.

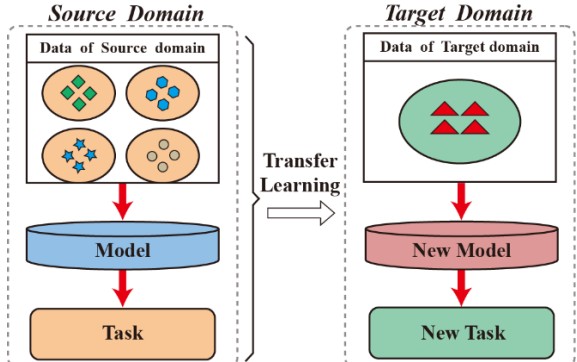

**Figure 1.** Schematic diagram of Transfer Learning.

Suppose a source domain with $n$ labels $D_s = \{x_{si}, y_{si}\}, i = 1, 2, 3...n$, and an unlabeled target domain $D_t = \{x_{tj}\}, j = 1, 2, 3...m$, where $x_{si}, x_{tj}$ represent the sample set and $y_{si}$ represents the sample label. Suppose the feature spaces of the source domain and target domain are the same, $X_s = X_t$, and the corresponding category spaces are also the same, $Y_s = Y_t$. However, their marginal and conditional probability distributions are not the same, such that:

$$P_s(x_s) \neq P_t(x_t),\ P(y_s|x_s) \neq P(y_t|x_t) \tag{1}$$

The purpose of transfer learning was to learn a classifier $f$ from the data in the source domain, which enables this classifier to use $x_t$ to predict $y_t$.

### 2.3. Weighted Balanced Distribution Adaptation Algorithm

The difference in distribution between domains will lead to poor performance of the classifier in the target domain [17]. However, domain adaptation can reduce the differences

in marginal and conditional probability distributions. A commonly used domain adaptation method is the balanced distribution adaptive (BDA) method, which adaptively weighs the importance of marginal and conditional probability distributions. It is calculated as follows:

$$D(D_s, D_t) \approx (1 - \mu)D(P(x_s), P(x_t)) + \mu D(P(y_s|x_s), P(y_t|x_t)), \tag{2}$$

where the first term on the right side of the equation represents the edge probability distribution adaption and the last term represents the conditional probability distribution adaption. Moreover, $\mu \in [0, 1]$ represents the adaptive balance factor for the adaptation distribution. When $\mu \to 0$ represents a dissimilar data set, the marginal probability distribution is more dominant. When $\mu \to 1$ represents the data set similarity, the conditional distribution is more adaptive.

Maximum mean discrepancy (MMD) is the most frequently used metric in migration learning, it is used to measure the distance between two distributions in the reproducing Hilbert space, and is a kernel learning method [18]. We employ it to calculate the differences between the two different distributions in Equation (2) and to adaptively minimize these differences. So, the discrepancy between the source domain and the target domain can be expressed as:

$$
\begin{aligned}
D(D_s, D_t) \approx \quad & (1 - \mu)\left\| \frac{1}{n}\sum_{i=1}^{n} x_{si} - \frac{1}{m}\sum_{j=1}^{m} x_{tj} \right\|_H^2 \\
& + \mu \sum_{C=1}^{C} \left\| \frac{1}{n_C} \sum_{x_{si} \in D_s^{(C)}} x_{si} - \frac{1}{m_C} \sum_{x_{tj} \in D_t^{(C)}} x_{tj} \right\|_H^2 .
\end{aligned}
\tag{3}
$$

where $H$ represents the reproduction of Hilbert space, and $n$ and $m$ represent the number of samples in the source domain and target domain, respectively. The $C$ is the label of different classes, and the $D_{s/t}^{(C)}$ represents samples of the $C$-th class from the source domain and target domain, respectively. The $n_C$, $m_C$ represent the number of experiments of $D_s^{(C)}$ and $D_t^{(C)}$, respectively.

However, the test data of ESPs in real production are not labeled, which makes $P(y_t)$ impossible to obtain directly, thus making the conditional probability distribution $P(y_t|x_t)$ not easy to calculate. Therefore, in this paper, a weighted balanced distribution adaptation (W-BAD) was used to better approximate the conditional probability distribution adaptation for ESP fault identification experiments. According to the Bayes theorem [19], we can neglect to calculate $P(y_t)$, which uses $P(x_t|y_t)$ to approximate $P(y_t|x_t)$, where $\alpha_s$, $\alpha_t$ represent class priors for two different domains:

$$
\begin{aligned}
\| P(y_s|x_s) - P(y_t|x_t) \|_H^2 &= \left\| \frac{P(y_s)}{P(x_s)}P(x_s|y_s) - \frac{P(y_t)}{P(x_t)}P(x_t|y_t) \right\|_H^2 \\
&= \| \alpha_s P(x_s|y_s) - \alpha_t P(x_t|y_t) \|_H^2 .
\end{aligned}
\tag{4}
$$

Equation (3) can be rewritten as:

$$
\begin{aligned}
D(D_s, D_t) \quad \approx \quad & (1 - \mu)\left\| \frac{1}{n}\sum_{i=1}^{n} x_{si} - \frac{1}{m}\sum_{j=1}^{m} x_{tj} \right\|_H^2 \\
& + \mu \sum_{C=1}^{C} \left\| \frac{\sqrt{P(y_s^{(C)})}}{n_C} \sum_{x_{si} \in D_s^{(C)}} x_{si} - \frac{\sqrt{P(y_t^{(C)})}}{m_C} \sum_{x_{tj} \in D_t^{(C)}} x_{tj} \right\|_H^2 .
\end{aligned}
\tag{5}
$$

The $P(y_s^{(C)})$ and $P(y_t^{(C)})$ in Equation (5) represent the prior probabilities of each of the different classes in the source domain and target domain, respectively. In the case of imbalanced samples, each class has a different probability of being in the domain. The W-BDA assumed that $P(x_s)$ and $P(x_t)$ are invariant and used prior probabilities to directly

approximate the conditional distribution probabilities, which overcomes the inconvenience of computing scatter in the conditional distribution. Equation (2) can be rewritten as:

$$\min tr(A^T X((1-\mu)M_0 + \mu\sum_1^C W_C)X^T A) + \lambda\|A\|_F^2,$$
$$s.t. A^T X H X^T A = I, 0 \le \mu \le 1. \tag{6}$$

The former part in Equation (6) is the adaptation of the marginal distribution and conditional distribution with a balanced factor, and the latter part is the corresponding regularization. The $\lambda$ is the regularization parameter with $\|\cdot\|_F^2$. The $X$ represents the data matrix consisting of $x_s$ and $x_t$. The $I \in \mathbb{R}^{(n+m)\times(n+m)}$ represents a unit matrix and the $H$ is a central matrix. The $A$ is the transformation matrix for minimizing the maximum mean difference in the conditional distribution in the source and target domains, and $M_0$ and $W_C$ represent the maximum mean difference matrix and the weight matrix, respectively, calculated as follows:

$$(M_0)_{ij} = \begin{cases} \frac{1}{n^2}, & x_i, x_j \in D_s, \\ \frac{1}{m^2}, & x_i, x_j \in D_t, \\ -\frac{1}{m\cdot n}, & other - situations. \end{cases} \tag{7}$$

$$(W_C)_{ij} = \begin{cases} \frac{P(y_s^{(C)})}{n_C^2}, & x_i, x_j \in D_s^{(C)}, \\ \frac{P(y_t^{(C)})}{m_C^2}, & x_i, x_j \in D_t^{(C)}, \\ \frac{\sqrt{P(y_s^{(C)})}\cdot\sqrt{P(y_t^{(C)})}}{n_C\cdot m_C}, & \begin{cases} x_i \in D_s^{(C)}, x_j \in D_t^{(C)}, \\ x_i \in D_t^{(C)}, x_j \in D_s^{(C)}, \end{cases} \\ 0, & other - situations. \end{cases} \tag{8}$$

Finally, the Lagrange multiplier $\Phi = (\Phi_1, \Phi_2, \Phi_3, ..., \Phi_d)$ was introduced, and the optimal transformation matrix was solved.

$$L = (X((1-\mu)M_0 + \mu\sum_1^C W_C)X^T + \lambda I)A$$
$$= X H X^T A \Phi. \tag{9}$$

### 2.4. Data Analysis

The ESPs dataset was obtained from the China Offshore Oil Development and Production database. The dataset used for the experiments covers three different failures of ESPs, with each working condition described by 15 different features: daily liquid production (DLP), Wellhead temperature (WT), test water volume (TWV), water to gas ratio (WGR), pump current (PC), pump voltage (PV), oil pressure (OP), oil to gas ratio (OGR), test liquid volume (TLV), daily water production (DWP), daily gas production (DGP), daily oil production (DOP), water content (WC), test oil volume (TOV), and gas to oil ratio (GOR). The three different working conditions are pipe column leakage, overload pump stop, and underload pump stop, which contain both normal and abnormal data. The data information for each specific ESP is shown in Table 1.

**Table 1.** ESP data information.

| No. | Symbol | Variable Name (Unit) | No. | Symbol | Variable Name (Unit) |
|-----|--------|----------------------|-----|--------|----------------------|
| **1** | DLP | Daily liquid production (m$^3$/day) | **9** | TLV | Test liquid volume (t) |
| **2** | WT | Wellhead temperature (°C) | **10** | DWP | Daily water production (m$^3$/day) |
| **3** | TWV | Test water volume (m$^3$/day) | **11** | DGP | Daily gas production (m$^3$/day) |
| **4** | WGR | Water gas ratio (%) | **12** | DOP | Daily oil production (m$^3$/day) |
| **5** | PC | Pump current (A) | **13** | WC | Water content (wt%) |
| **6** | PV | Pump voltage (V) | **14** | TOV | Test oil volume (t) |
| **7** | OP | Oil pressure (kPa) | **15** | GOR | Gas-oil ratio (%) |
| **8** | OGR | Oil gas ratio (%) | | | |
| **Working Conditions** | | | **Working Status** | | |
| Column leakage (1) | | | Lines break, disconnect, wear, and corrode, resulting in leaks. | | |
| Overload pump stopping (2) | | | Overload current setting is not reasonable, the motor is impaired, the pump is mixed with impurities, etc. Overload shutdown occurs. | | |
| Underload pump stopping (3) | | | Underload current setting is not reasonable, pump or separator shaft is broken due to insufficient fluid supply from the ground. | | |

## 3. Algorithm Design

### 3.1. Method Flow

The method based on the integration of unsupervised and multi-source transfer learning was divided into the following steps, and its flow is shown in Figure 2.

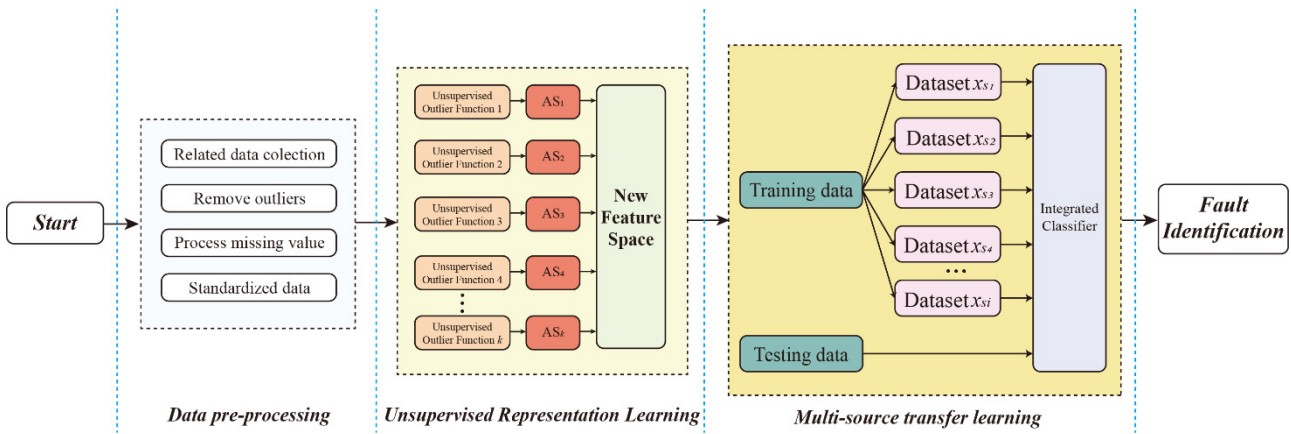

**Figure 2.** Unsupervised and multi-source transfer learning integration flow chart.

Step 1: Data pre-processing of the collected data, including data cleaning, missing value filling, outlier processing, normalization, etc.
Step 2: Inputting data into multiple unsupervised methods to construct a new feature space and enhance the information representative of a minority of fault samples.
Step 3: Unduplicated random sampling of the training data while ensuring that a minority of faulty samples are adequately sampled.
Step 4: Multiple training to obtain reliable base classifiers.

Step 5: Multiple recognition results are integrated through multiple base classifiers to obtain the final recognition results.

### 3.2. Phase I: Data Pre-Processing

The ESP history database contains a large number of process variables that reflect actual production, some of which may have missing values or outliers. In order to improve the data quality of the monitoring model and the accuracy of the model [20], the data need to be pre-processed and finally normalized to eliminate the effect of the magnitude between the data by scaling the data to $[0, 1]$.

$$x_{norm} = \frac{(x_i - x_{min})}{(x_{max} - x_{min})},$$

(10)

where $x_{norm}$ represents the result of the normalization of the variable, and $x_{max}$ and $x_{min}$ represent the maximum and minimum values of the $i$-th variable, respectively.

### 3.3. Phase 2: Unsupervised Representation Learning

Unsupervised anomaly scoring (AS) can be treated as a form of raw data learning features to be used to enhance the space of raw features [21]. The AS function is used to define a mapping function $\Psi(\cdot)$ in the feature space $X \in \mathbb{R}^{n \times d}$, and each different mapping function outputs a new vector $\Psi_i(X) \in \mathbb{R}^{n \times 1}$, which uses AS to describe the degree of anomalies in the data. A transformation function matrix $\Psi = [\Psi_1, \Psi_2, \Psi_3, ..., \Psi_k]$ is constructed by combining $k$ anomaly scoring functions, which is used to generate the AS matrix on the feature space $X$. These extracted AS are combined with the original feature space to obtain the new feature space $X_{new} = [X, \Psi(X)] \in \mathbb{R}^{n \times q}, q = d + k$, which improves overall abnormality recognition. The process is shown in Figure 3.

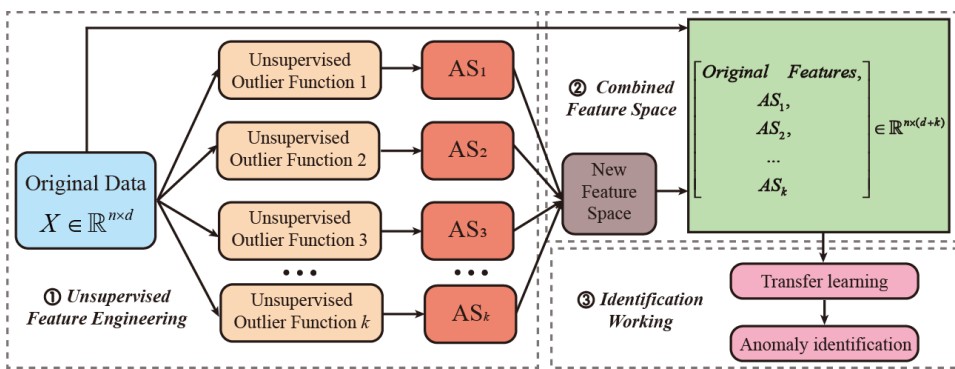

**Figure 3.** Description of unsupervised feature engineering.

In this study, five base anomaly scoring functions were used: k-nearest neighbor (KNN) [22], local outlier factor (LOF) [23], one-class SVM [24], isolation forests (IFs) [25] and autoencoder (AE) [26]. To further increase the diversity, the parameters of each model were different, and we set multiple parameters for each model, and selected from each model a few sets of AS for better performance. We used the Euclidean distance [27] as the distance metric in KNN, and the range of $k$ was defined as $\{1, 2, 3, 4, ..., 100\}$. For the range of $k$ in the LOF, it was also defined as $\{1, 2, 3, 4, ..., 100\}$. For one-class SVM, the kernel function used the radial basis function (RBF). For IF, the number of basic evaluators was fixed in the range $\{5, 10, 20, 40, 80, 160, 200, 250\}$. For AE, different network layers and depths were used, and the sigmoid function was chosen as the activation function.

### 3.4. Phase 3: Multi-Source Transfer Learning

Simply increasing the original feature space is not enough to improve fault diagnosis accuracy. The minority samples must be sufficiently trained to build an effective training model. The traditional weak classifiers have difficulty in identifying minority samples,

and the identification results tend to be biased in favor of majority samples [28]. We proposed a transfer learning model with a multi-source training set, which is shown in Figure 4. The training data were randomly sampled without repetition while ensuring that the minority samples were drawn in an adequate manner. In total, we obtained a source domain $X_s = \{x_{s1}, x_{s2}, x_{s3}, ..., x_{si}\}$ consisting of $i$ sample training sets. This ensured that the minority class sample was fully utilized and increased the weight of the minority sample. Then multiple base weak learners were trained based on the samples from the source domain, and so on until the number of base learners reached a pre-specified value $T$. The error rate of the base learner was calculated using Equation (11), and the one with the smallest error rate was selected as the base classifier. The $T$ base learner-identified results were assembled into a strong classifier discriminant to obtain the final identification results (Equation (12)), where the II represents the characteristic function, indicating that it is 1 when the working condition in the brackets holds, and 0 otherwise.

$$\varepsilon = \frac{1}{|X_s|} \sum_{x \in X_s} \sum_{t=1}^{T} (h_t(x_{si}) \neq y_{si})/T. \tag{11}$$

$$R(x) = argmax \sum_{t=1}^{T} \text{II}(h_t(x_{si}) = y_{si}). \tag{12}$$

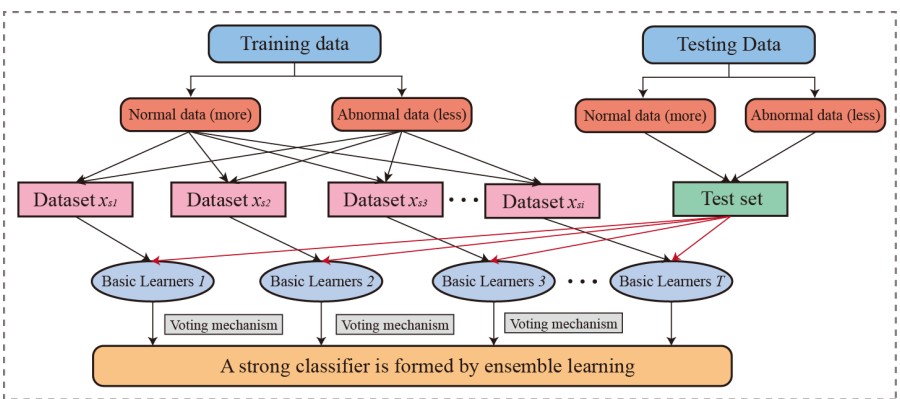

**Figure 4.** The multi-source transfer learning process.

To clarify the balance factor $\mu$, A-distance [29] is used as the basic measure in this paper. A-distance is defined as a linear classifier built to distinguish the hinge loss of two data domains. Defining $\varepsilon(h)$ as a linear classifier $h$ distinguishes two domains: the error between the source domain $D_s$ and the target domain $D_t$. A-distance is calculated as:

$$d_A(D_s, D_t) = 2(1 - 2\varepsilon(h)). \tag{13}$$

We define $d_M$ as the A-distance of the edge distribution, and $d_C$ represents the conditional distribution distances corresponding to the categories, calculated as $d_C = d_A(D_s^{(C)}, D_t^{(C)})$. The $\mu$ can be calculated by the following equation:

$$\hat{\mu} = 1 - \frac{d_M}{d_M + \sum_1^C d_C}. \tag{14}$$

## 4. Classifier Performance Evaluation Method

In the imbalanced data classification problem, positive class samples are the ones given more attention [30], and the prediction accuracy for the whole dataset does not fully reflect the good or bad performance of the imbalanced learning method. Let us assume that an imbalance problem has 100 samples, of which 10 samples are negative class and 90 samples are positive. If the classifier predicts all the samples as a positive class, its accuracy is still

as high as 90%. However, from a practical point of view, such a classifier does not make any sense, because the minority samples are all misclassified, and in a practical application scenario, this classifier is invalid. This also happens in ESP fault identification. If the faulty sample is misjudged as a normal sample, it is not possible to stop production and maintenance in time, which leads to more serious accidents. To objectively and adequately measure the classification of unbalanced data, a confusion matrix needs to be constructed to determine the relevant metrics. The confusion matrix is shown in Table 2.

**Table 2.** Confusion matrix.

|  | **Predicted for Positive Class** | **Forecast for Negative Class** |
|---|---|---|
| True for positive class | TP | FN |
| True for negative class | FP | TN |

The recall is the ratio of the number of samples correctly classified into positive classes to all positive classes and can be used to measure the performance of the classifier in identifying positive classes. It is calculated as follows:

$$\text{Recall} = \frac{\text{TP}}{\text{TP} + \text{FN}}. \tag{15}$$

TNR indicates how many of the total negative class samples are predicted to be negative, and can be used to measure the ability of the classifier to identify negative class samples. It is calculated as follows:

$$\text{TNR} = \frac{\text{TN}}{\text{TN} + \text{FP}}. \tag{16}$$

Precision indicates how many samples are correctly classified out of all samples judged to be positive classes. It is calculated as follows:

$$\text{Precision} = \frac{\text{TP}}{\text{TP} + \text{FP}}. \tag{17}$$

The F1-score is an inverse relationship between Precision and Recall, and it is one-sided to judge the classifier based on one of the two metrics. The combined metric is calculated as follows:

$$\text{F1} - \text{score} = \frac{2 \cdot \text{Recall} \cdot \text{Precision}}{\text{Recall} + \text{Precision}}. \tag{18}$$

G-mean is also an evaluation metric that is often used to measure the classification performance of an unbalanced dataset as a whole. Among them, TPR reflects the classifier's ability to recognize a few classes, while TNR is a reflection of the classifier's ability to recognize most classes. It is calculated as follows:

$$\text{G} - \text{mean} = \sqrt{\text{TPR} \times \text{TNR}}. \tag{19}$$

## 5. Experimental Analysis

### 5.1. Comparison of Classification Effects with and without Extraction of New Features

The experiments in this section are designed to demonstrate the impact of learning methods through unsupervised feature extraction. To ensure the accuracy of the experiment and eliminate random interference, we randomly select 50 different experimental data sets according to different imbalance ratios and then input each ratio into different methods to calculate their G-mean and F1-score, and finally derive their mean values. Considering the methods for electric submersible pump fault identification in existing studies, two commonly used methods, the XGB-based method and the SVM-based method, were selected here. The former is an integrated learning method, and the latter has a very

powerful effect in small sample learning. In the experiment, we randomly draw different experimental data sets from working condition 1 according to the above requirements. The experimental results are shown in Figures 5 and 6.

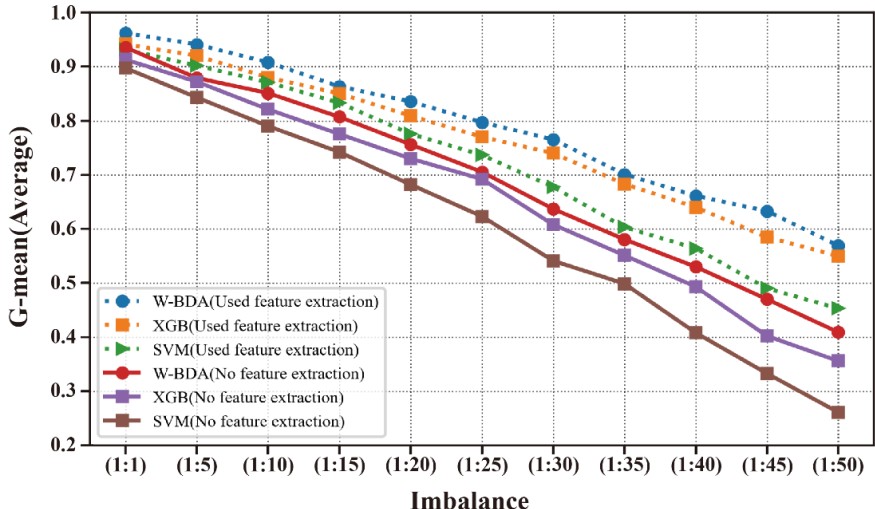

**Figure 5.** The G-mean of different methods with or without feature extraction.

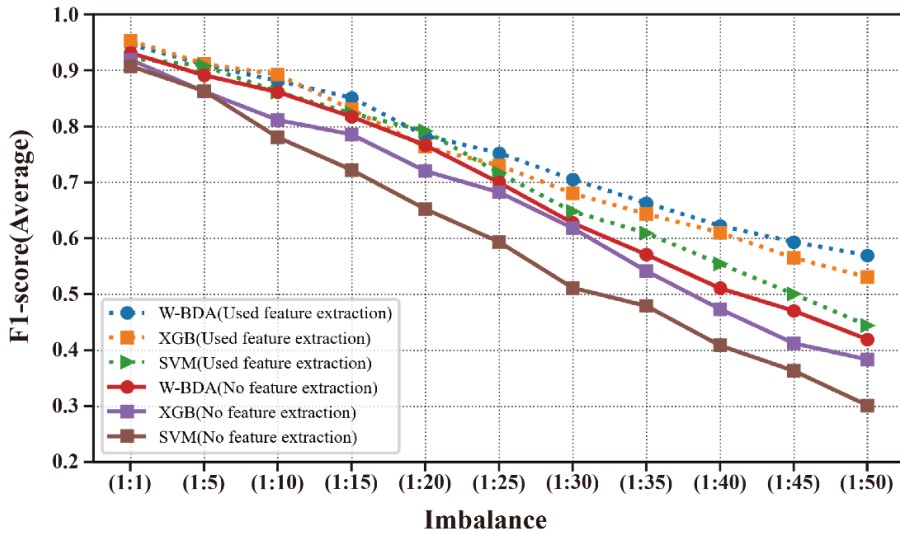

**Figure 6.** The F1-score of different methods with or without feature extraction.

As can be seen in Figures 5 and 6, the G-mean and F1-score of either method showed a decline when the imbalance ratio gradually increased. However, methods with the use of unsupervised feature extraction decline relatively slowly, which shows that the new feature space can add more data representation in combination with the original features, which facilitates the information capture of the method, and thus the recognition effect will be better than the method without the use of unsupervised feature extraction. However, the overall recognition was poor and fluctuated. Because the information of a small amount of anomalous data is submerged in a large amount of normal data, it is more difficult for the model to learn the representation information related to the anomalous data, which also shows that the imbalance of the data greatly interferes with the actual recognition effect.

## 5.2. Effect of the Learning Framework

The superiority of the UMTLA integration method proposed in this paper is demonstrated by comparing it with other imbalanced learning methods. A specific description of the 11 classification methods used for comparison is presented in Table 3. We divided

the compared methods into four main categories: classifiers without sampling, classifiers with sampling, integrated classification without sampling, and integrated classification with sampling.

**Table 3.** Eleven comparative classification methods.

| | |
|---|---|
| Classifier without Sampling | SVM_R_W: SVM with Gaussian function and imbalance weight |
| | DT: Decision Trees [31] |
| | MLP: Multi-Layer perceptron [32] |
| Classifiers with Sampling | R + SVM_R_W: Random sampling [33] + SVM_R_W |
| | S + SVM_R_W: SMOTE [34] + SVM_R_W |
| | G + SVM_R_W: Gaussian Oversampling [35] + SVM_R_W |
| Integrated Classification without Sampling | RF: Random Forest [36] |
| | XGB: XGBoost [37] |
| | ADA: ADAboost [38] |
| Integrated Classification with Sampling | R + XGB |
| | REMDD |

The classifiers without sampling are mainly the three commonly used simple classifiers, including support vector machines, decision trees, and a multi-layer perceptron. Classifiers with sampling are the combination of the SVM with different sampling methods. The main integrated methods without sampling are the random forest with the decision tree base classifier, the XGBoost with CART as the base classifier, and the ADAboost with the SVM as the base classifier. The integration methods with sampling are the REMDD (resampling ensemble model based on data distribution [39]) and the RUS + XGBoost. The imbalance ratios of 200:1, 100:1, 50:1, 30:1, 20:1, 15:1, 10:1, 5:1, 2:1, and 1:1 were set as the experimental data. In addition, the low standard deviation of the G-mean values obtained by cross-validation implies that the method has strong computational stability. The experimental results are shown in Tables 4 and 5.

**Table 4.** Average G-mean and standard deviation of 12 methods on 10 datasets.

| Methods | 1:1 | 1:2 | 1:5 | 1:10 | 1:15 | 1:20 | 1:30 | 1:50 | 1:100 | 1:200 |
|---|---|---|---|---|---|---|---|---|---|---|
| SVM_R_W | 0.9190 ± 0.03 | 0.9025 ± 0.01 | 0.8576 ± 0.00 | 0.8027 ± 0.01 | 0.6771 ± 0.00 | 0.6355 ± 0.00 | 0.5691 ± 0.01 | 0.5013 ± 0.20 | 0.3781 ± 0.01 | 0.1341 ± 0.00 |
| DT | 0.9201 ± 0.00 | 0.9165 ± 0.00 | 0.8211 ± 0.00 | 0.7557 ± 0.00 | 0.6013 ± 0.00 | 0.5261 ± 0.04 | 0.4971 ± 0.00 | 0.4043 ± 0.00 | 0.2239 ± 0.01 | 0.0963 ± 0.00 |
| MLP | 0.9284 ± 0.01 | 0.8433 ± 0.01 | 0.7287 ± 0.02 | 0.6562 ± 0.02 | 0.4870 ± 0.01 | 0.3873 ± 0.12 | 0.2431 ± 0.10 | 0.1141 ± 0.02 | 0.0692 ± 0.00 | 0.0072 ± 0.00 |
| RUS + SVM_R_W | 0.9112 ± 0.02 | 0.8452 ± 0.03 | 0.8461 ± 0.01 | 0.8233 ± 0.03 | 0.8081 ± 0.07 | 0.7433 ± 0.10 | 0.6523 ± 0.02 | 0.5508 ± 0.06 | 0.3066 ± 0.01 | 0.2036 ± 0.01 |
| S + SVM_R_W | 0.9338 ± 0.01 | 0.9008 ± 0.02 | 0.8601 ± 0.04 | 0.7930 ± 0.02 | 0.7330 ± 0.00 | 0.6964 ± 0.30 | 0.6156 ± 0.07 | 0.4364 ± 0.00 | 0.3272 ± 0.01 | 0.2513 ± 0.01 |
| G + SVM_R_W | 0.9510 ± 0.03 | 0.9364 ± 0.00 | 0.8272 ± 0.00 | 0.7772 ± 0.00 | 0.7073 ± 0.02 | 0.6234 ± 0.00 | 0.5672 ± 0.00 | 0.4655 ± 0.01 | 0.3770 ± 0.03 | 0.2041 ± 0.02 |
| RF | 0.9577 ± 0.00 | 0.9071 ± 0.00 | 0.8270 ± 0.11 | 0.7531 ± 0.14 | 0.6547 ± 0.00 | 0.5394 ± 0.01 | 0.4394 ± 0.01 | 0.3301 ± 0.00 | 0.2211 ± 0.03 | 0.1255 ± 0.05 |
| XGB | 0.9605 ± 0.03 | 0.9231 ± 0.01 | 0.9377 ± 0.00 | 0.8621 ± 0.08 | 0.7741 ± 0.01 | 0.6431 ± 0.00 | 0.5943 ± 0.10 | 0.3961 ± 0.01 | 0.2545 ± 0.00 | 0.1305 ± 0.01 |
| ADA | 0.9431 ± 0.08 | 0.9156 ± 0.00 | 0.8514 ± 0.05 | 0.8062 ± 0.12 | 0.7641 ± 0.07 | 0.7034 ± 0.02 | 0.6253 ± 0.00 | 0.4201 ± 0.02 | 0.3774 ± 0.09 | 0.3152 ± 0.04 |
| RUS +XGB | 0.9645 ± 0.00 | 0.9471 ± 0.00 | 0.9067 ± 0.00 | 0.8481 ± 0.01 | 0.8041 ± 0.07 | 0.7331 ± 0.05 | 0.5243 ± 0.00 | 0.4261 ± 0.00 | 0.3145 ± 0.04 | 0.2505 ± 0.01 |
| REMDD | 0.9205 ± 0.04 | 0.8723 ± 0.02 | 0.7737 ± 0.07 | 0.7551 ± 0.16 | 0.6741 ± 0.29 | 0.6031 ± 0.17 | 0.5143 ± 0.27 | 0.4561 ± 0.01 | 0.3345 ± 0.04 | 0.2805 ± 0.09 |
| UMTLA | 0.9821 ± 0.01 | 0.9644 ± 0.00 | 0.9245 ± 0.01 | 0.9121 ± 0.01 | 0.8841 ± 0.02 | 0.8554 ± 0.01 | 0.8104 ± 0.03 | 0.7715 ± 0.01 | 0.7443 ± 0.02 | 0.7042 ± 0.01 |

**Table 5.** Average F1-score and standard deviation of 12 methods on 10 datasets.

| Methods | 1:1 | 1:2 | 1:5 | 1:10 | 1:15 | 1:20 | 1:30 | 1:50 | 1:100 | 1:200 |
|---|---|---|---|---|---|---|---|---|---|---|
| SVM_R_W | 0.9212 ± 0.01 | 0.8933 ± 0.02 | 0.8311 ± 0.02 | 0.7559 ± 0.03 | 0.6912 ± 0.00 | 0.6051 ± 0.01 | 0.5334 ± 0.04 | 0.4231 ± 0.12 | 0.3581 ± 0.01 | 0.1401 ± 0.01 |
| DT | 0.9014 ± 0.00 | 0.9554 ± 0.01 | 0.7812 ± 0.01 | 0.7256 ± 0.01 | 0.6634 ± 0.02 | 0.5022 ± 0.01 | 0.5312 ± 0.02 | 0.4166 ± 0.01 | 0.2934 ± 0.01 | 0.1776 ± 0.20 |
| MLP | 0.8814 ± 0.00 | 0.8754 ± 0.03 | 0.7731 ± 0.01 | 0.7056 ± 0.01 | 0.6015 ± 0.01 | 0.5054 ± 0.02 | 0.4069 ± 0.11 | 0.3054 ± 0.23 | 0.2014 ± 0.05 | 0.1267 ± 0.00 |
| RUS + SVM_R_W | 0.9256 ± 0.01 | 0.9025 ± 0.01 | 0.8434 ± 0.01 | 0.7912 ± 0.01 | 0.7045 ± 0.01 | 0.6394 ± 0.01 | 0.5667 ± 0.11 | 0.4012 ± 0.01 | 0.3256 ± 0.08 | 0.2107 ± 0.02 |
| S + SVM_R_W | 0.9512 ± 0.02 | 0.8933 ± 0.02 | 0.8512 ± 0.01 | 0.7723 ± 0.03 | 0.7212 ± 0.00 | 0.6721 ± 0.02 | 0.5878 ± 0.01 | 0.4261 ± 0.04 | 0.3298 ± 0.02 | 0.2314 ± 0.01 |
| G + SVM_R_W | 0.9472 ± 0.02 | 0.9237 ± 0.01 | 0.8617 ± 0.03 | 0.8021 ± 0.03 | 0.7091 ± 0.02 | 0.6464 ± 0.12 | 0.5961 ± 0.02 | 0.4651 ± 0.03 | 0.3312 ± 0.01 | 0.2001 ± 0.02 |
| RF | 0.9225 ± 0.03 | 0.8731 ± 0.02 | 0.7937 ± 0.01 | 0.7451 ± 0.10 | 0.6641 ± 0.01 | 0.5931 ± 0.02 | 0.4742 ± 0.04 | 0.3961 ± 0.02 | 0.3045 ± 0.01 | 0.2205 ± 0.30 |
| XGB | 0.9675 ± 0.01 | 0.9014 ± 0.03 | 0.8943 ± 0.01 | 0.8347 ± 0.01 | 0.7671 ± 0.02 | 0.6746 ± 0.01 | 0.5859 ± 0.07 | 0.4673 ± 0.02 | 0.3611 ± 0.02 | 0.2522 ± 0.02 |
| ADA | 0.9431 ± 0.08 | 0.9156 ± 0.00 | 0.8514 ± 0.05 | 0.8062 ± 0.12 | 0.7641 ± 0.07 | 0.7034 ± 0.02 | 0.6253 ± 0.00 | 0.4201 ± 0.02 | 0.3774 ± 0.09 | 0.3152 ± 0.04 |
| RUS +XGB | 0.9633 ± 0.02 | 0.9179 ± 0.01 | 0.8672 ± 0.00 | 0.7613 ± 0.01 | 0.6821 ± 0.02 | 0.5483 ± 0.02 | 0.4638 ± 0.03 | 0.3732 ± 0.01 | 0.3215 ± 0.01 | 0.2621 ± 0.02 |
| REMDD | 0.9616 ± 0.01 | 0.9055 ± 0.02 | 0.8261 ± 0.01 | 0.7611 ± 0.13 | 0.6358 ± 0.21 | 0.5387 ± 0.14 | 0.4768 ± 0.20 | 0.4023 ± 0.05 | 0.3559 ± 0.01 | 0.2832 ± 0.04 |
| UMTLA | 0.9738 ± 0.02 | 0.9474 ± 0.01 | 0.9038 ± 0.00 | 0.8734 ± 0.01 | 0.8451 ± 0.01 | 0.8105 ± 0.03 | 0.7504 ± 0.02 | 0.7344 ± 0.02 | 0.6912 ± 0.02 | 0.6491 ± 0.01 |

We can see from Tables 4 and 5 that the G-mean and F1-score perform relatively well in the first few cases in which the proportion of data imbalance is small for all the methods. However, the recognition effect of traditional single-learner methods (SVM, DT, etc.) decreases significantly when the imbalance ratio reached 1:15, which is due to the large gap between the number of normal samples and abnormal samples, and the information of a few abnormal samples is covered by the information of most normal samples, resulting in poor learning of data feature information related to abnormal samples by the model and a higher false recognition rate. For integrated learning, the performance of classifiers was improved by overlaying and integrating single classifiers, but the feature information of minority fault samples still cannot be learned effectively when the degree of data imbalance is large. Using the undersampling method resulted in the loss of more counterexamples, which caused the loss of important information and led to the fact that the classifier recognition effect would be more inclined to the majority fault samples, and the G-mean and F1-score decreased significantly. On the contrary, when the method of oversampling was used, since the imbalance of the data was filled by making a difference between the minority faulty samples, it was equivalent to adding many new spurious samples, and their recognition effect was affected by the synthetic samples, which also had a poor recognition effect on the minority fault samples. The method proposed in this paper, based on the full utilization of minority samples in the data processing stage, used random sampling to form a source domain consisting of multiple training sets to ensure adequate training of minority samples, increase the weights of minority fault samples, and then improve the performance of the model by integrating multiple base classifiers, effectively solving the problem of data imbalance. The results of the experiments with the data of working conditions 2 and 3 are shown in Figures 7–10 which also illustrate the effectiveness of the method very well.

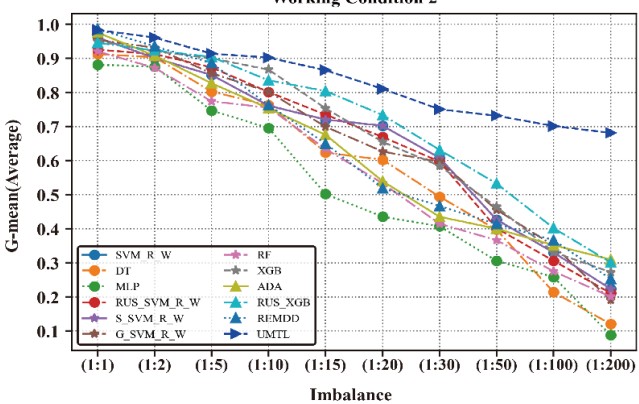

**Figure 7.** G-mean of 12 methods on 10 different imbalances of data on working condition 2.

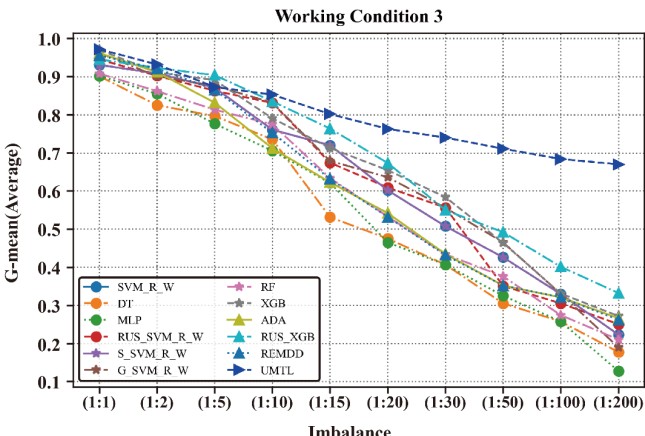

**Figure 8.** G-mean of 12 methods on 10 different imbalances of data on working condition 3.

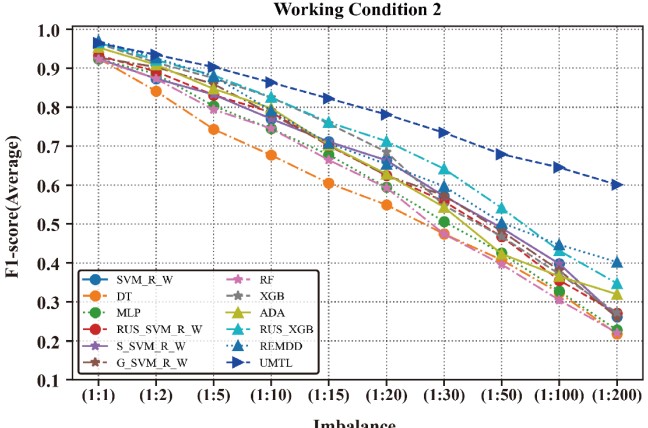

**Figure 9.** F1-score of 12 methods on 10 different imbalances of data on working condition 2.

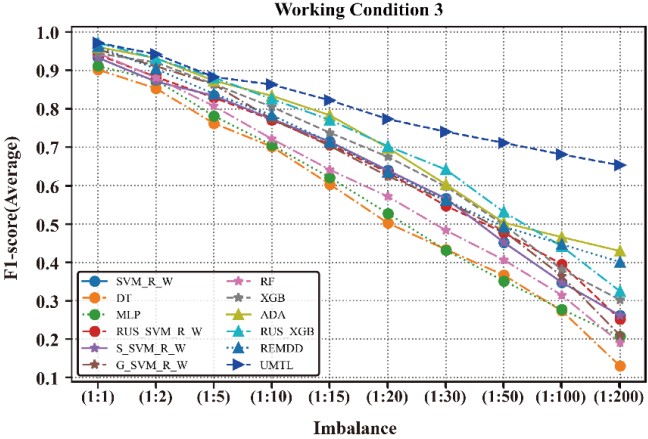

**Figure 10.** F1-score of 12 methods on 10 different imbalances of data on working condition 3.

To further test the effectiveness of the UMTLA on different distributions of the ESPs' data, experiments were conducted by using data from different working conditions as training and test sets, with condition 2 as the training data and working conditions 3 and 1 as the test data, respectively. By comparing the SVM, XGB, and the TrAdaboost [40], the parameters of the SVM and XGB were kept as before. The auxiliary samples of TrAdaboost were selected from the source domain, and the number of iterations $N = 25$. The experimental results are shown in Figures 11 and 12.

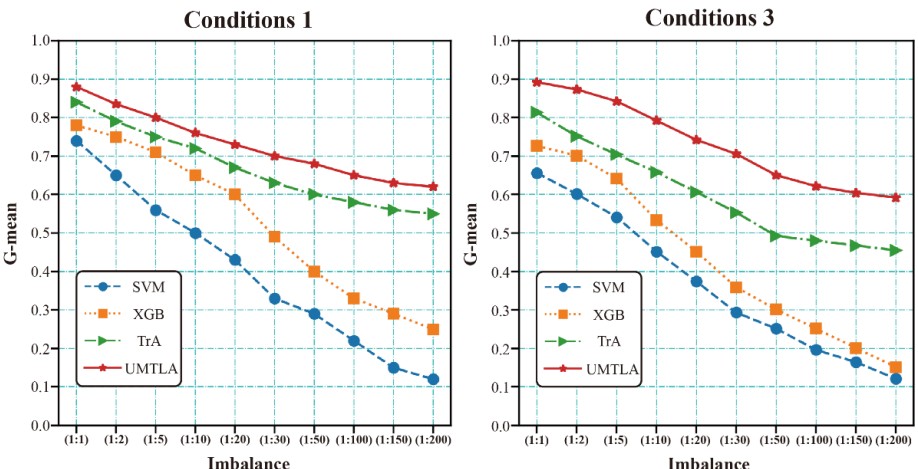

**Figure 11.** Comparison G-mean of imbalanced sample learning methods.

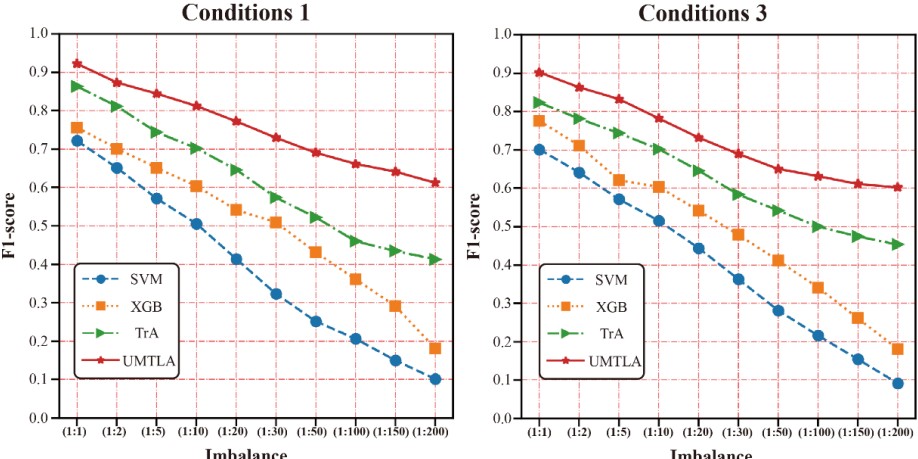

**Figure 12.** Comparison F1-score of imbalanced sample learning methods.

The observation plot reveals that the G-mean and F1-score decrease with the increase of the imbalance ratio. The SVM and XGB had the largest declines, because the training data (working condition 2) and test data (working conditions 3 and 1) have a large state span and different data distribution. Traditional machine learning was easily limited by the working condition of independent identical distribution, and the imbalance ratio increased to be unable to learn the information of minority fault samples effectively, leading to a serious decline in the identification effect. Because TrAdaBoost has the same weight update method for both samples, and its auxiliary data are vulnerable to the interference of unbalanced samples, making it more difficult to identify, the performance is average in both working conditions' data. In this paper, we proposed UMTLA to fully train minority fault samples fully random multiple sampling with multiple source inputs, while expanding the learning of minority fault samples through the weighted integration of multiple classifiers. Furthermore, using the source domain and target domain prior probabilities to approximate the conditional probabilities and adding different weights to the two types of samples, it made the G-mean and F1-score of UMTLA not less than 0.60, which indicates a relatively good recognition ability for minority fault samples.

## 6. Discussion

Compared with other ESP fault identification methods, the innovation of the method proposed in this paper is mainly reflected in the following two aspects.

As for the data features, we addressed the difficulty of identifying fault samples in electric submersible pump data due to the lack of expressiveness of sample features. We

extract the anomaly score (AS) as a richer representation of the data by combining several unsupervised learning methods, which enhances the learning ability of the model for a few classes of data and improves the overall model's identification of fault samples.

In terms of fault samples, a framework of multi-source transfer learning is proposed to ensure that the minority samples can be utilized to maximize, thus improving the perception and weight of the minority class samples. Moreover, a weighted balanced adaptive approach (W-BDA) is introduced to reduce the variation between the source and target domains.

Although the method proposed in this paper has good results in dealing with ESP fault identification in the case of imbalanced data, it suffered from the problem of having a long running time due to a large number of unsupervised methods and the large number of base classifiers in the transfer learning framework. We will try more data as well as methods to improve the running speed and feasibility of the model in the future, which is the main research direction in the future.

## 7. Conclusions

We proposed a fault identification method based on the integration of unsupervised and multi-source transfer learning for the ESPs' fault. We have used the characteristics of unsupervised learning to extract new data information representations to enhance the data representation; the method of randomly selected samples can better learn the information of a few fault samples, and the class prior was used to more accurately approximate the probability of conditional probability distribution to improve the recognition of samples under different distribution conditions. Experiments showed that the method has a high accuracy in identifying unbalanced data and can effectively cope with the problems of inaccurate fault identification caused by imbalanced data problems. It is very interesting and innovative to explore the application of ESP fault identification under the problem of imbalanced data using methods, such as unsupervised learning and transfer learning. We will continue to study this topic in the future. We also hope that this paper can provide a new idea in the field of ESP faults.

**Author Contributions:** Conceptualization, P.Y. and L.W.; methodology, P.Y. and S.L.; software, P.Y. and L.W.; validation, P.Y. and J.C.; formal analysis, P.Y.; investigation, P.Y. and L.W.; resources, L.W.; data curation, P.Y. and J.C.; writing—original draft preparation, P.Y.; writing—review and editing, P.Y. and S.L.; visualization, P.Y. and L.W.; supervision, S.L.; project administration, S.L.; funding acquisition, S.L. All authors have read and agreed to the published version of the manuscript.

**Funding:** This research was funded by the Southern Marine Science and Engineering Guangdong Laboratory, Zhanjiang (Grant No. ZJW-2019-04).

**Institutional Review Board Statement:** Not applicable.

**Informed Consent Statement:** Informed consent was obtained from all patients involved in the study.

**Data Availability Statement:** Data can be made available upon reasonable request.

**Conflicts of Interest:** The authors declare no conflict of interest.

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
