# Peer review of "Fault Identification of Electric Submersible Pumps Based on Unsupervised and Multi-Source Transfer Learning Integration"

_sustainability, doi:10.3390/su14169870_

Round 1
Reviewer 1 Report
Paper is about improving the fault identification of electric submersible pumps. The topic is worth of investigating and lies in the scope of the journal, because the sustainability can be also treated as a concern about minimizing the faults. The benefits of improved fault system are clear and obvious.
Paper is well written and the scientific soundness is good.
I have some major comments that has to be taken into account by authors.
1. ESP in the title should be described because this abbreviation can be not readable for Sustainability Readers. The scope of the journal is very wide. It is not enough to present the expansion of the abbreviations in abstract.
2. Please highlight in the literature review the differences between previous papers and your paper. Please clearly indicate the knowledge gap and prove that it is a really not analyzed area of the field. Please indicate new approach / new methods in a comparison to the existing investigations.
3. Please divide conclusions section in two parts: discussion and conclusions. In discussion, please provide more information about simplifications and their possible effect on the results and on the conclusions (separately). Please make a discussion about the possible practical application of the work, especially if it is possible, please do an additional simple analysis of the benefits reached in the exemplary practical application in short term and long-term application. Please discuss future work.
4. In conclusions, please add only short statements that contains QUANTITATIVE findings of the work and please highlight the great meaning of the work and new / sufficient contribution to the field. It is very important to present here only main findings in short sentences and with the numbers not only words and description. Words and description should be placed in the discussion section.
Author Response
Dear Reviewer.
Thank you for acknowledging our manuscript and for your comments on our manuscript. These comments are valuable and helpful in revising and improving our paper, as well as providing important guidance for our research. We revise the manuscript according to your requirements. Please find attached our answers to the questions you have asked. The questions are in black font, our answers are in red font, and the revisions to our manuscript are in blue font.
Thanks very much for your attention and consideration.
Best regards,
Yours sincerely,
Sheng Li

Reviewer 2 Report
The authors of this manuscript propose a multisource unsupervised transfer learning methodology to identify faults in electric submersible pump (ESPs). This approach is necessary to identify defects due to the lack of expressiveness of the sample features and the small number of unevenly distributed data samples.
The proposed approach has been validated by experiments using ESP production data from South China Sea Petroleum.
The manuscript is generally well written, but I suggest that the authors make the following major revisions before it is accepted for publication.
1) The abstract needs to be rewritten as it is really unclear. I understand that the proportion of majority and minority samples is seriously unbalanced, but the authors do not explain at all why they are interested in minority samples. Furthermore, to prove the feasibility and effectiveness of the proposed method, the authors must quantify the results obtained.
2) In the general introduction, although the authors' contributions are clearly expressed, the research questions are not defined.
3) Sections 3 and 4 need to be reworked. I propose that the authors define a section "3. Materials and Methods" in which they can define a methodology with a general scheme to help the reader understand the ins and outs of the proposed work and only then, they can explain the implemented algorithm, as well as the classification method.
4) Another major point: the authors do not explain well how their work is part of a sustainable approach. As a result, we can ask ourselves the question of the choice of the journal!
5) The authors need to further justify the use of the G-mean metric. Is it sufficient to discriminate minority samples? What other metrics could have been used and if so, why did the authors not choose them?
6) The authors do not explain at all the interest of equations 15 to 17.
7) Concerning the experimental results, figures 4 and 5 lack explanations. What should we retain? What will these results be used for?
8) It is imperative to define a "Discussion" section in order to step back and discuss the experimental results obtained. This discussion must provide clear answers to the research questions previously defined.
9) The general conclusion is far too brief. It is necessary to recall the context of the study, to synthesize the main ins and outs of the previously defined research questions, to explain the limits of the proposed methodology and finally, to give research perspectives.
10) As far as the formatting of the manuscript is concerned, the equations take up too much space. It looks like a copy/paste of equations whereas it is imperative to use an equation editor. All equations must be announced in the body of the text. Some figures need to be reworked because they are unreadable (figures 4 to 7). The bibliographical references can be completed by recent articles published in the journals of the MDPI group.
Author Response

(The authors gave the same response as above.)

Round 2
Reviewer 2 Report
The authors responded to each of my comments with precision and seriousness.
I thank them for that!
I therefore propose that the manuscript be published as is.